# Tensile Properties of Additively Manufactured Thermoplastic Composites Reinforced with Chopped Carbon Fibre

**DOI:** 10.3390/ma15124224

**Published:** 2022-06-14

**Authors:** Jaroslav Majko, Milan Vaško, Marián Handrik, Milan Sága

**Affiliations:** Faculty of Mechanical Engineering, Department of Applied Mechanics, University of Žilina, Univerzitná 8215/1, 010 26 Žilina, Slovakia; milan.vasko@fstroj.uniza.sk (M.V.); marian.handrik@fstroj.uniza.sk (M.H.); milan.saga@fstroj.uniza.sk (M.S.)

**Keywords:** tensile test, experimental measurement, nylon reinforced with chopped carbon fibre, carbon fibre, material deposition strategy

## Abstract

3D printing allows controlled deposition of composite components, which the user defines by the modification of the printing parameters. The article demonstrates that all observed printing parameters (infill type, infill orientation) influence the tensile test results of nylon reinforced with chopped carbon fiber. The highest tensile strength obtains specimens with the maximum number of walls around the circumference. The plastic region of the tensile diagram differs significantly with the change of material orientation in the structure, as the specimens with material deposited 45/−45 to the load axis have four times greater tensile strains and 20% higher tensile stresses than 0/90. The assessment of results reveals the significant difference between deformations at break and permanent deformations. In addition, the permanent lateral strain reaches up to 20%. Finally, the article consists of a brief assessment of the printing parameters (printing time, weight) of individual series. The future modelling in FEA software requires additional experiments to verify the viscoelastic properties of the material.

## 1. Introduction

Additive manufacturing is a modern production method based on the successive deposition of material layer upon layer to create a designed object [1]. From an engineer’s point of view, its working principle significantly differs from the currently prevailing technologies based on material removal (milling, grinding, turning). The principle of material addition provides advantages such as, for example, the production of variously shaped objects [2,3,4], and significantly contributes to the reduction of material consumption and production cost [5]. In consideration of limited staffing requirements, it represents a milestone in the interests of rich countries to return production from countries characteristic of low labour costs. For this reason, there is a tendency to support technology research within the so-called Fourth Industrial Revolution [6].

Although 3D printing has been undergoing a significant increase of interest only recently, the current state is the result of long-term development that began in the 1980s. The fundamental technologies of additive manufacturing are:Extrusion—works on the principle of material deposition by nozzle on the printing bed. The significant advantages of the method are its simplicity, low cost, and controlled deposition of material into the composite structure at precisely defined locations [7];Vat photopolymerization—uses ultraviolet light to trigger the polymerization process of the photopolymer. The production of FRP composites is possible by mixing the fibre with the resin or by dispersing the fibre on the resin surface. The fibres are oriented in different ways [8,9];Powder bed fusion—based on the action of thermal energy on the powder dispersed on the workplace. Due to technological limitations, both the matrix and the reinforcement are in powder form [10];Sheet lamination—works on the principle of joining a sheet of material to form a part. The method involves additive and subtractive techniques, adding a sheet of material and then cutting it off with a laser to the desired shape [11].

Each of the mentioned methods is appropriate for a particular type of material. Subsequent development of the methods allows the usage of additive technologies in other areas [12,13,14]. From the point of view of composite production, the first two technologies are of interest since their operational principle enables the production of fibre-reinforced composites with a precise deposition of the components in the structure [15,16]. In the case of thermoplastic composites printing, the most widespread is the Fused Filament Fabrication (FFF) method (described in more detail in Section 2.1), which allows the printing of composites reinforced with chopped or long filament without significant modifications of the technology.

Composites are materials that present a physical combination of at least two components at the macroscopic level. In general, the properties of composites surpass the properties of the materials of which they are composed [17,18]. In most cases, the mentioned properties are stiffness, strength, weight, fatigue strength, corrosion resistance, etc. [19]. Typical examples of composites are concrete, wood, or bone tissue [20,21,22,23]. The fundamental component of the composite is a matrix fulfilling the function of a binder. The reinforcing function performs reinforcement in the form of fibres, flakes, or dispersed particles [24]. With the expansion of plastics into the market, fibre reinforced polymer composites have begun to be applied in practice [25]. Depending on the fibre length, the polymers can be reinforced with long or short fibres [26,27,28,29], however, composites reinforced with long fibres achieve better mechanical properties [30].

Currently, additive manufacturing enables the printing of fibre-reinforced composites. The primary advantages of printing composites using the FFF method are the production speed and the ability to control the arrangement of materials in the structure by adjusting the printing parameters. This paper aims to observe the effect of these parameters on the tensile properties of a composite composed of nylon reinforced with chopped carbon material. To date, published studies have focused on tensile testing of the laminate, primarily investigating the mechanical properties of specimens with triangular and rectangular infill types. The authors published a study [31], which compared the influence of lamina thickness, infill type and infill density on ultimate tensile force and material consumption. Naranjo-Lozada et al. [32] claimed that chopped carbon fibre reinforced nylon laminates achieved higher tensile strength values than pure nylon laminates in all observed infill patterns by approximately 15 to 20%. One of the primary aims was to establish the influence of infill patterns (rectangular and triangular) and infill pattern densities (10 and 70%) on tensile test results. According to the results, both parameters affected the tensile properties. The highest yield strength, tensile strength, and Young’s modulus achieved specimens with triangular infill pattern and 70% infill density. The authors came up with the explanation that the triangular infill pattern provides the highest values of strength due to a higher proportion of filament with parallel orientation to the load axis. The differences between the yield strength and tensile strength were from 25 to 40%. The strain values vary between 0.1 and 0.2 mm/mm. Yasa et al. [33] discussed the printing accuracy and tensile properties of specimens made of chopped fibre-reinforced nylon with various infill patterns. The results demonstrated the suitability of the printing technology for a dimensional tolerance of more than 0.1 mm. A comparison of the observed infill strategies showed that the highest yield strength reached the specimens filled with rectangular infill patterns, which presents a contrast to the results of the previous study. The highest values of strength and Young’s modulus in elasticity obtained specimens with rectangular infill and XZ printing plane. The measured tensile strength values were up to 60 MPa with relative strains at break up to 30% and Young’s modulus of elasticity of 3 GPa.

An overview of already published research studies on the topic identified that their focus is solely on specimens with rectangular or triangular infill patterns with varying infill density. However, none of the studies contain information on how the solid infill affects the tensile properties. The motivation of this paper is to follow up the findings described in the previous paragraph with a series of tensile tests performed on the same material with a solid infill type raster with varying material orientation in the structure for the following reasons:Following the argumentation in article [32] regarding the higher strength of triangular infill due to a higher proportion of filament oriented in the load axis direction, it follows that it is necessary to identify the influence of the orientation of the printed material concerning the load axis on the achieved tensile properties;During the preparation process in the slicing software, the user may change the object orientation on the work plane (e.g., the object does not fit due to its length). In the case of some printers, this modification means changing the orientation of the material placement, which may inadvertently affect the resulting tensile properties of the printed composite by the user;Throughout the lifetime of the printed object, it may experience loading in different directions regarding the orientation of the material in the structure;The material under investigation performs the function of the matrix of the CFRP composites investigated in the various articles [34,35,36], and it is essential to know the properties of its constituents to properly understand its behaviour during loading;Modelling of composites in FEA software inevitably requires the identification of the material characteristics of the individual components.

A series of these experiments followed the modification of other printing parameters (such as the number of walls), which from preliminary observations show the effect on tensile properties and printing time.

Therefore, the last sections contain a brief study of the printing parameters modification effect on the weight of printed objects and on the printing time. In the future, the results from these tensile tests will help with the proposal for a suitable FRTP composite prediction strategy in FEA software.

## 2. Materials and Methods

### 2.1. Manufacturing Process

Printing of the specimens was carried out on a Markforged MarkTwo printer [37], whose working principle, based on the FFF method, is as follows: the extruder pushes the material wound on the spool into the printer head, which moves in the XY plane along precisely specified paths. At the same time, the material in the printing head is melted and deposited at a specified location. After the layer completion, the printer continues with the subsequent layers until the whole object is created [38,39,40]. The advantage of the method is the precise deposition of material at specified locations, allowing the user to adjust the filament deposition pattern as required (Figure 1).

In the case of the MarkTwo printer, the orientation of the laminas in the laminate is the perpendicular lay-up sequence. The user can modify the object orientation on the printing desk, but the laminas are always deposited alternately perpendicular to each other (Figure 2, left and middle). Measurements have shown that the width of one reinforced nylon filament is 0.4 mm. Inside the filament are chopped carbon fibres (Figure 2, right), whose arrangement in the structure is not uniform [41]. The length of chopped carbon fibres ranges from 140 to 210 µm. The fibre diameter in published scientific studies varies from 7 to 10 µm [32,33].

The laminae thickness can be 0.1, 0.125, or 0.2 mm depending on the parameters set by the printer user. In addition, the user could select the appropriate infill strategy (rectangular, triangular, hexagonal, solid) and infill density.

### 2.2. Material

The specimens were printed from nylon reinforced with chopped carbon fibre (trade-mark Onyx). The manufacturer supplies this material as fibre wound on a spool. The mechanical properties of nylon reinforced with chopped carbon fibre are listed in Table 1.

### 2.3. Specimens

The proposed specimen shape was according to ASTM D638-14 (Figure 3a). The specimens (Figure 3b) were printed by a Markforged MarkTwo printer, whose working principle is based on the FFF method. The thickness of the specimens was 3 mm. This value is in the ±0.2 mm tolerance specified by the given standard. The experimental measurement confirmed the suitability of the selected specimen shape since the experience has shown that most of the specimen failure occurs in the predefined critical area. The reason is that the narrowest cross-section in the specimen middle expected the highest stresses in the region.

The specimen preparation consisted in creating the model in CAD software according to the specified geometry and saving it as the stl file format. Subsequently, the file was imported into slicing software, which allows modifying the printing parameters. The summary of printing parameters applied in the experiment is listed in Table 2.

The selected lamina thickness was 0.1 mm. The article focused only on specimens with solid infill type and infill density of 100%. Due to the lamina thickness, the total number of laminas was 30. The primary aim was to observe the influence of selected printing parameters (infill orientation, number of walls, number of floor/roof layers) on tensile properties. A schematic sketch of what the mentioned printing parameters represent is shown in Figure 4.

Infill orientation in the structure is a parameter customized by the user in the slicing software. The user chooses from the following infill patterns: hexagonal, triangular, gyroid, rectangular, and solid. The program permits the rotation of specimens on an imaginary printing desk to allow the user to fit large sized specimens on it (Figure 5). However, in the case of some printers, rotation of the specimen results in a change of the infill orientation in the structure, which may affect the mechanical properties of the printed object in practice (for example, reduction of tensile strength), and the user of the printer may not be aware of this.

It is important to note that in the case of the MarkTwo printer, the layers are deposited alternately perpendicular to each other (Figure 6).

The second observed parameter was the number of walls around the specimen circumference. The purpose of the walls is to maintain the shape of objects under the influence of loading. Preliminary data and observations pointed out that the application of walls leads to a longer printing time. The slicing software allows adjusting the number of walls from 0 to 15 (Figure 7).

The final observed parameter is the number of roof and floor layers (Figure 4). According to the manufacturer, this parameter has the following functions: protection of the structure from water and increasing the aesthetic level of the surfaces. Slicing software allows users to modify the number of these layers from 0 to 10. The application of these layers will create a sandwich consisting of (Figure 4):Solid/triangular/hexagonal/rectangular infill, and;Roof/floor infill.

Preliminary observations show that this parameter affects the printing time and the density of material deposition in the lamina. Examination showed that floor/roof layers in the narrowest part of the specimen consisted of 33 filaments. In contrast with solid infill, these layers have one extra filament (Figure 8).

## 3. Tensile Test Results

The tensile test results of the selected specimen types are shown in Table 3. Each series consisted of five specimens. The test was performed on INSTRON 5985 tensile machine.

According to the preliminary results, the yielding force and maximum force depend on the following parameters (in descending order):Number of walls;Number of roof and floor layers, and;Infill orientation.

The initial analysis of the measured results shows that the highest tensile forces reached specimens with a filament deposited around the specimen circumference. The measured maximum force was 1566.22 N. In contrast, the minimal value of ultimate force was achieved by specimens without walls.

In addition, the number of floor and roof layers has a significant effect on the tensile force values. The presented results confirmed the direct proportionality between the number of layers and measured tensile force. The difference in the tensile force of specimen series η (specimens without floor and roof layers) and specimen series ε (specimens with maximum roof and floor layers) is 21%.

Finally, the infill orientation is another parameter that affects the measured values of tensile force. The maximal tensile forces were reached by specimens with the infill orientation 45/−45 degrees. On the other hand, usage of 0/90 infill orientation led to the lowest values of tensile forces.

The initial analysis of the measured force results indicates significant differences between the individual specimen series. Therefore, the subsequent sections contain a more in-depth analysis of the measured results. In addition, the last sections analyse the effect of printing parameters on the specimens’ weight and print time.

## 4. Influence of the Printing Parameters on the Tensile Properties

The tensile test results in Table 3 indicate a significant effect of the material arrangement on maximum force and yielding force values. For a better illustration of the results, the authors prepared diagrams, which help with a deeper analysis of the measured values. The calculations of tensile stresses consisted of the fraction of the measured force to the area of the narrowest part of the specimens without considering any printing imperfections.

### 4.1. Orientation of the Infill Pattern to the Loading Axis

The first observed parameter was the orientation of the material pattern in the structure to the loading force. We assessed four fundamental patterns: 45/−45 (alpha), 0/90 (beta), 15/−75 (gamma) and 30/−60 (delta). The force-displacement curves are shown in Figure 9.

#### 4.1.1. Yield Strength and Ultimate Strength

Firstly, the authors assessed the measured values of yield stress (Figure 10). The lowest values achieved specimens of the delta series. On the other hand, the specimens with the infill orientation 15/−75 degrees (gamma) reached the highest yield stress at 15.79 MPa. The maximum difference between the series was approximately 8%.

From the generated diagrams, we could observe the effect of material orientation on the specimen stiffness along the loading. Related to this fact, the pattern at 0/90 degrees transmits higher loads in the elastic region with smaller deformations.

With the transition to the plastic area, the specimens’ behaviour begins to vary significantly. The specimens of alpha series with infill orientation 45/−45 degrees reach a tensile strength of 30 MPa. It is substantially more than in the case of beta specimens with 0/90 infill orientation, which achieved 20% less tensile strength. This observation proves that the orientation of the material affects the tensile strength.

The authors compared the tensile strength of the alpha specimens (infill orientation 45/−45) with the results in Refs. [31,32]. The measured values of tensile strength in the mentioned studies on similar specimens were 17 and 37 MPa, respectively. The former had an infill density of 70%, which may explain the more significant differences. The latter had the same infill density but a higher number of walls and a lamina thickness of 0.125 mm. However, the measured value of the alpha series is in accordance with the data in Table 1 provided by the printer manufacturer.

#### 4.1.2. Strain

In addition, the authors observed significant differences in measured elongation and relative strains in the direction of the load axis. Table 4 comprises the values of elongation at the yield limit, elongation at failure, and permanent set.

Especially in the case of the 45/−45 infill orientation, the authors observed significant ductility of the laminate. Usage of other infill orientations led to a considerable decrease in ductility. The lowest values of specimen elongation were recorded in the beta series. At the end of the measurement, the authors performed measurements of the permanent set in the direction of the load axis, which indicated significant contractions of the laminate. The length of laminates with 45/−45 infill orientation dropped to 40% of elongation at the fracture. The contraction of specimens with 0/90 infill orientation was even bigger since the length of the specimens decreased by 73%. Therefore, the permanent set of the beta specimens is approximately equal to elongation at the yield point. Infill pattern 45/−45 did not reveal any deformations, only in the normal direction. The additional measurement showed a permanent narrowing of the alpha specimens by 3 mm and significant specimens twisting (Figure 11a). Assessment of the 0/90 infill pattern did not identify any specimen twisting (Figure 11b).

The elongation values allow the determination of normal strains at yield point and normal strains at fracture (Figure 12). In all cases, the normal strain within the elastic region was approximately at the same values, from 3 to 5%.

In the case of plastic deformation, the differences between the specimen series are more noticeable. The specimens from the alpha series have the largest plastic region with normal strain at fracture 44.2%. The modification of laminas orientation from 45/−45 to 0/90 degrees is connected with a considerable decrease of normal strain.

The measured strain at the fracture of alpha specimens was compared with the results in Refs. [31,32]. The comparison showed considerable differences, since alpha specimens have two times larger strain values. The explanation could be the difference between the compared specimens specified in Section 4.1.1. On the other hand, the material data given in Table 1 show strain at the fracture of almost 60%, which is close to the measured values for the alpha specimens.

The effects of laminas orientation are also confirmed by images from the failure region of the selected two series of specimens in Figure 13. The crack path is different in both cases. In the case of 0/90 lamina orientation, the crack path is perpendicular to the direction of the loading axis (Figure 13a). In case of 45/−45 infill pattern, the vast majority of the crack path is in the 45/−45 degree to the loading axis (Figure 13b). The sections along the filament are interrupted by very short perpendiculars intersecting this interfilamentous fracture. The presence of walls around the specimen circumference affected the trajectory of the crack in the edge area since the wall filament orientation is parallel to the loading axis and also different to infill pattern orientation. In all series, the crack trajectory depends on the infill pattern orientation to loading.

In order to identify the behaviour of the material during loading, cross-sections of the ruptured alpha and beta specimens were investigated using SEM (Figure 14 and Figure 15). A comparison of Figure 14 and Figure 15 show a significant difference between the observed series in terms of the proportion of voids in the structure. In the case of the 0/90 material deposition, the void was between each filament within a single lamina. The presence of voids contributes to the limited connection between the filaments and the laminas.

A higher magnification of the images (Figure 14b,c and Figure 15b,c) allowed us to observe the effect of tensile loading on the individual composite phases—the broken carbon fibres and the extensively stretched nylon filaments. In addition, the images (especially Figure 15d) show a laminate failure process where, in the case of the 0/90 laminate, the filament was broken in the laminas oriented in the direction of the load axis. In the case of laminas oriented perpendicular to the loading direction, the fracture propagated between the filaments, indicating a lower strength between them. A similar laminate failure mode was observed for the 45/−45 laminates (Figure 14d).

The influence of infill pattern orientation on the significant difference of tensile diagrams could explain the total number of laminas that efficiently transfer the loading. In the case of 45/−45 infill pattern orientation, all layers participate in the unified distribution of the stress in the structure. The assessment of results identified that the increase of angle between the lamina orientation and the loading axis led to a decrease of normal strength and normal elongation in the plastic region. The extreme presents the specimens with 0/90 infill orientation, where half of the laminas are oriented perpendicular to the load and thus the load transfer through these layers is minimal, i.e., only half of the laminate layers effectively transfer the loading.

The properties of composites highly depend on the individual components and shape of their chemical chain. Nylon is a viscoelastic polymeric material characteristic of high ductility. On the contrary, carbon fibre achieves high strength values due to the alignment of atoms along the axis of the fibre. The main disadvantage of carbon fibre is its high brittleness. This knowledge allows making a presumption that transmission of loading from filaments oriented parallel to load axis to fibres oriented perpendicular to load axis results in shear loading of filaments reinforced with chopped carbon fibres and subsequent break of fibres. In the 45/−45 raster orientation, atoms of reinforcing carbon fibre are better oriented to the direction of the loading axis, and therefore the carbon fibre transmits higher loads. However, in both cases of raster orientation, the failure mechanism is similar. Comparison of the normal strains between specimens made of pure nylon [31] and the carbon fibre reinforced nylon indicate that the presence of carbon fibre affects the strain values. Additionally, both components influence the subsequent shortening of the specimens after the experiment. The viscoelastic behaviour of nylon results in the tendency to return the specimen into its original shape. The presence of chopped carbon fibre also magnifies the intensity of the tendency. The FE calculation considered in the future will depend on a suitable choice of material model, which depend on the measurements focused on the directional dependence of the mechanical properties and viscoelasticity of the material.

### 4.2. Number of Walls

The subsequently observed parameter was the number of walls around the specimen circumference. Within the experiment plan, the authors chose the following number of walls: 0, 2, and 15. A preliminary assessment of the tensile test results identified the effect of the parameter on the shape of the tensile diagram. The potential differences in the tensile diagram included various degrees of steepness of the curves in the elastic region and different sizes of the elastic and plastic region (Figure 16).

#### 4.2.1. Yield Strength and Ultimate Strength

A comparison of the results between the individual series showed an apparent influence of the number of walls around the specimen circumference on the yield strength (Figure 17). The lowest values achieved specimens without walls. The increase in the number of walls to the maximum will increase the yield strength by more than 50%.

The assessment of the tensile strength showed the similarity with yield strength results, i.e., the specimens with the maximal number of walls reached the highest tensile stress values. In general, the achieved tensile strength by the zeta series is the highest of all observed specimens. Subsequent analysis of results revealed that the relationship between the number of walls and the yield strength or tensile strength is linear, where one additional wall increases the specimen strength by 1.25-fold. The specimens consisted of 30 identical laminas, whereas each lamina in the narrowest part consisted of 30 filaments with parallel orientation to the loading axis. Based on these parameters, the tensile strength of one wall filament is 56.12 MPa.

#### 4.2.2. Strains

The tensile test results showed a significant difference in the elongation of the specimen series with the various number of walls (Table 5). In general, the decrease in the number of walls led to the increase of measured elastic displacements. Therefore, the lowest values of elastic elongation were obtained by specimens with the maximal number of walls.

In comparison with the elastic region, the plastic region demonstrates the opposite results. The primary difference comprises the two following observations. Firstly, the measured values confirm the direct proportion between the number of walls and the plastic displacements. Secondly, the displacements in the plastic region are seven times greater than in the elastic region. In addition, the authors performed the measurements of the permanent set. The specimens with 15 walls were contracted by 72%. Since all specimens with various numbers of walls reported similar values of the permanent set, the authors exclude the influence of the walls on the contraction degree. In addition, permanent transverse displacement was of the half-millimetre.

As in the previous subsection, the values of elastic strains ranged from 2 to 5%. Specimens without walls had elastic strains approximately the same as specimens with a material orientation of 30/60 degrees (Figure 18).

The differences between the plastic strains are less than in the case of specimens with modified infill orientation. The highest plastic strains had specimens with the maximum number of walls. The differences of plastic strains between specimens with solely a few walls were not significant. In general, the influence of the number of walls on the deformation of the specimen is relatively smaller compared to infill orientation.

The crack propagation path was similar to specimens with a material orientation at an angle of 0/90 in the previous section. Nevertheless, the locations of crack initiation were various (Figure 19). Specimens without walls ruptured in the middle of the narrowed section. On the contrary, the specimens with walls have the crack located at the beginning of the curved section. The reason is an occurrence of a stress concentrator in this part of the specimen.

The benefit and simultaneously disadvantage of the strategy with the maximum number of walls around the specimen circumference is the unidirectional deposition of material in all laminas, which ensures the orientation of filaments with chopped carbon fibre parallel to the load axis. Carbon fibres provide excellent properties and significantly contribute to the superior properties of the laminate in this direction. The disadvantage is that these properties are limited only in that orientation, i.e., the composites are characteristic of the directional dependence of the mechanical properties.

Finally, for the purpose of assessing the changes in microstructure during loading, SEM images of the fractured cross-section of the zeta series specimen were realised (Figure 20).

In contrast to the arrangement of the 0/90 in Figure 15, a lower occurrence of voids between the filaments was observed. The individual filaments were distinguished even at 300× magnification (for the alpha and beta series, 100× magnification was sufficient). In addition, the higher magnification allows observing many voids in the filament, which are most likely the result of the chopped fibres being pulled out of the nylon.

### 4.3. Number of Floor and Roof Layers

The last observed parameter was the number of roof and floor layers. The printing process of these laminas is typical of a higher number of filaments and slower printing speed than in standard layers (Section 2.3). As a result of these modifications, the printing time of laminates has been extended (see Section 6). The primary aim was to observe the effect of floor and roof layers on the tensile properties (Figure 21).

#### 4.3.1. Yield Strength and Ultimate Strength

According to the tensile test results, the number of the floor and roof layers has an influence on values of the yield strength (Figure 22). The increasing number of these laminas led to a higher elastic stiffness in the normal direction and maximum yield stress values of approximately 15.6 MPa. The difference in yield stress between the minimum and maximum number of floor and roof layers was 13%.

A comparison of the tensile strength force results of the eta and epsilon series show the increasing differences in the plastic region. The specimens with the maximum number of floor and roof layers had a maximum force of 1440 N, which corresponds to the ultimate strength 34 MPa. In the case of specimens without the layers, the ultimate strength was approximately 25% lower. The relationship between strength and the number of roof and floor laminas is appropriately expressed by a quadratic function, i.e., the increase of strength weakens with the increasing number of the laminas.

#### 4.3.2. Strain

The tensile diagrams of the epsilon and the eta series showed differences in the average values of the elongations (Table 6). Specimens without floor and roof layers (eta series) had elongations at a yield point bigger by 20% than the specimens with maximum number of floor and roof layers (epsilon series). On the other hand, the maximal value of elongation at failure was reached by the epsilon series specimens, but the difference was 10%.

The permanent sets in both extreme modifications were almost the same. Therefore, the number of floor and roof layers does not significantly affect the subsequent contraction of specimens. The measured values of permanent transverse displacements achieved 2 mm.

Based on these observations, the authors plotted the relationship between the calculated strains and the number of specified layers (Figure 23).

The maximal elastic strains were reported by the specimens with zero number of floor and roof layers. As the number of these layers increased, the elastic deformations decreased exponentially. The range elastic strain values ranged from 4 to 5%.

The comparison of strains at fracture revealed that the maximal strains achieved specimens with two floor and roof layers (44.2%). Subsequently, an increasing number of the layers did not lead to the considerable growth of strain values. The differences between the series did not exceed 10%.

The crack path was oriented 45 degrees to the direction of the loading (Figure 24). The exception presented at the edges with walls, where the crack orientation was perpendicular to the loading axis. The failure mechanism reported similarity with specimens 45/-45 degrees material orientation described in Section 4.1. Therefore, the number of floor and roof layers has not influenced the crack propagation path.

## 5. Influence of Printing Parameters on Specimen Weight

The measured specimens weight ranged from 8.3 to 8.72 g depending on the printing parameter modification. The maximum weight was reached by the epsilon series with 10 floor/roof layers. The zeta specimens with 15 walls around the circumference also showed a relatively higher weight. The difference between the lightest and heaviest specimen series is approximately 5%. Therefore, none of the assessed deposition strategies significantly influence the specimens’ weight and material consumption. Comparison of various infill orientations showed that specimens with an infill orientation of 0/90 degrees had the lowest weight, but at the same time, this configuration also required the minimal value of force to reach the ultimate strength. Other infill orientation strategies had a slightly higher weight (approximately 2%), but the ultimate strength was significantly higher (Table 7).

Subsequently, the authors assessed the effect of the walls, roof, and floor layers on the specimen strength (Table 8).

The assessment of results showed that the number of walls had the greatest effect on the specimen weight. However, the effect on the specimen weight is relatively negligible in comparison with the influence of the parameter on the specimen strength. Similar conclusions pertain in the case of the roof and floor layers. According to the results, the best printing strategy is the maximal number of walls and roof/floor layers.

## 6. Influence of Printing Parameters on Printing Time

Finally, the printing time was the last assessed parameter. In comparison with observations in previous section, the differences between specimen series in the case of printing time are significantly higher. The printing of the zeta specimen required 105 min, which absolutely present the longest printing time (Figure 25). The graph expressing the printing time of the individual series revealed the direct proportion between the printing time and particular printing parameters (walls, roof, and floor layers). The decrease of the number of walls or roof and floor layers to the minimum reduced the printing time by 30%.

Among the specimen series with modified infill orientation, the differences in printing time are negligible. On the other hand, these series indicated greater differences in tensile strength. Therefore, the alpha specimens with infill angle 45/−45 are the best choice. The interesting observation is the negligible difference in printing time between alpha and epsilon series. Based on that, the authors recommend the usage of a maximal number of floor and roof layers as possible in the design process. The reduction of these layers led to only a 10% decrease in printing time, while the ultimate strength decreased by 20% (Table 9).

The decrease of the number of walls is accompanied by a significant reduction of printing time and an even greater decrease in ultimate strength. The relationship between the number of walls and ultimate strength is defined by a direct proportion. Finally, the authors performed a comparison between the number of walls and the infill angle of 45/−45 degrees. The 45/−45 infill deposition strategy led to a 26.5% reduction in printing time and a 25% decrease in ultimate strength. In general, the infill angle 45/−45 with the maximal number of roof and floor appears as the most appropriate printing strategy. On the contrary, the maximal number of walls present a better choice on the request of the highest possible strength regardless of the printing time. The assessed series represent extreme configurations, and, in practice, the user can choose a compromise, for example, infill orientation 45/−45 degrees with two or three walls around the specimen.

## 7. Conclusions

The results of tensile test measurements performed on chopped carbon fibre reinforced laminates made by 3D printing proved the sensitivity of the results to the orientation of the material in the structure. The assessment reveals the following observations:The highest yield strength was achieved by the specimens with the filament oriented at an angle of 15/−75, whereas the differences between the individual series were not significant (up to 10%). The highest strains at yield limit achieved specimens with infill type 45/−45. However, the differences between series were approximately 30%;After the transition to the plastic region, the differences between the specimens rapidly increased. The specimens with 45/−45 infill orientation had many times higher strain at fracture than the 0/90 infill pattern;The usage of 45/−45 infill orientation led to the highest tensile strength (30 MPa);The measurements identified a considerable difference between the strains at the failure and the permanent set. Depending on infill orientation, the contraction rate of the individual specimen series varied from 40 to 70%;The minimal contractions were measured in the case of specimens with the 45/−45 infill orientation. At the same time, this infill orientation reported the largest permanent set in the transverse direction (23%).

Subsequent assessment of other parameters has shown their influence on the tensile properties. In the case of the tensile strength, the influence can be sorted in a descending order: the number of walls > the number of floor and roof layers > infill orientation.

The results showed that an increase in the number of walls and the number of roof and floor layers affected the yield strength and tensile strength of specimens. In terms of strains, the differences were not as significant as in the case of infill orientation. The specimens with the maximum number of walls had the highest values in both the longitudinal and transverse directions. In addition, the specimens with the maximal number of walls reached the maximal tensile strength (40 MPa) of all observed specimen series.

Finally, the authors assessed how these printing parameters affected specimen weight and printing time:The difference between specimen series was negligible (5%);The number of walls had the greatest influence on the printing time. In comparison with other series, the printing time of the specimens was at least a quarter higher.

Therefore, the most suitable series are specimens with the infill orientation of 45/−45 degrees. If the designer is not limited by printing time, the number of walls also appear as the appropriate strategy.

## Figures and Tables

**Figure 1 materials-15-04224-f001:**
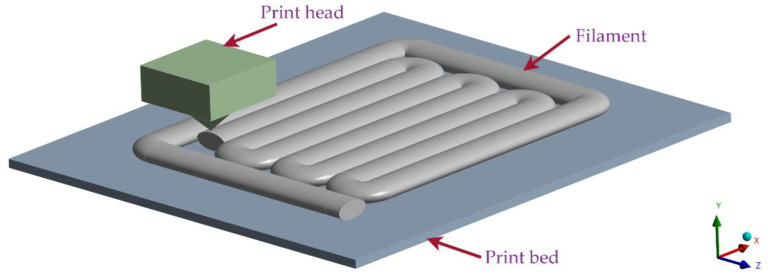
Fundamental scheme of fused filament fabrication printer. The controlled deposition of material into structure allows printing of various infill pattern types.

**Figure 2 materials-15-04224-f002:**
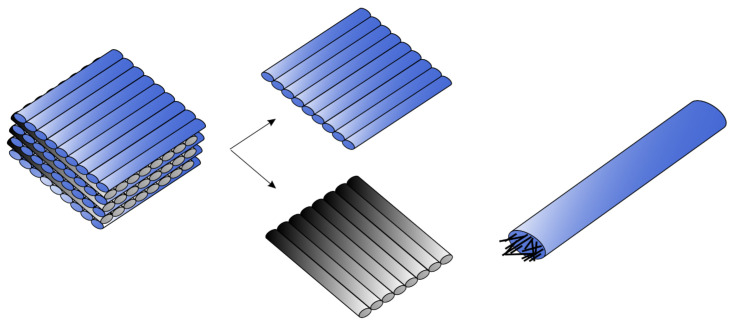
Perpendicular ply sequence (**left**); laminas orientation (**middle**); filament reinforced with chopped carbon fibre (**right**).

**Figure 3 materials-15-04224-f003:**
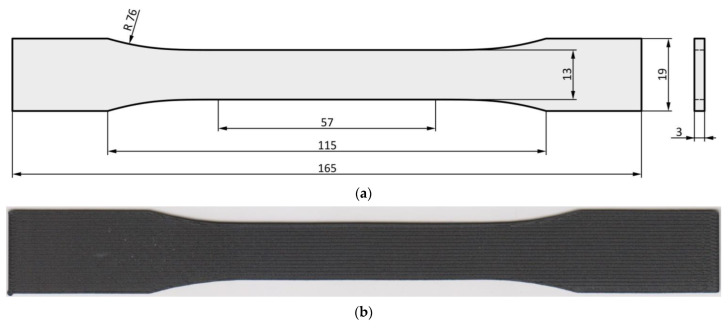
Specimen shape: (**a**) dimensioned drawing according to ASTM D638-14; (**b**) photo of a printed specimen.

**Figure 4 materials-15-04224-f004:**
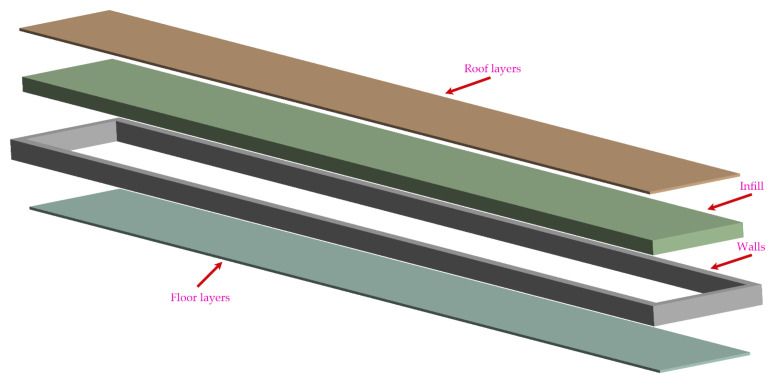
Schematic sketch of printed parts.

**Figure 5 materials-15-04224-f005:**
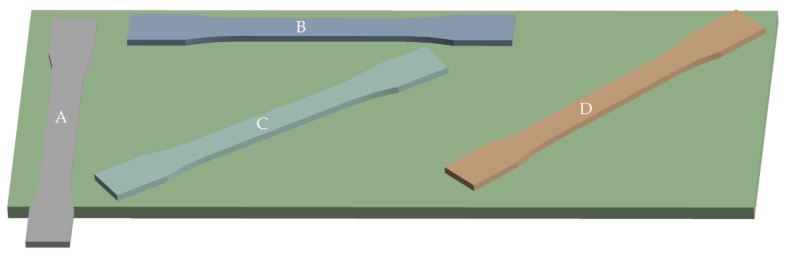
Specimen A exceeds the printing bed. However, the rotation of the specimen to a specified degree (B—0 degrees; C—30 degrees; D—45 degrees) allows fitting the specimen on the printing bed and successful printing.

**Figure 6 materials-15-04224-f006:**
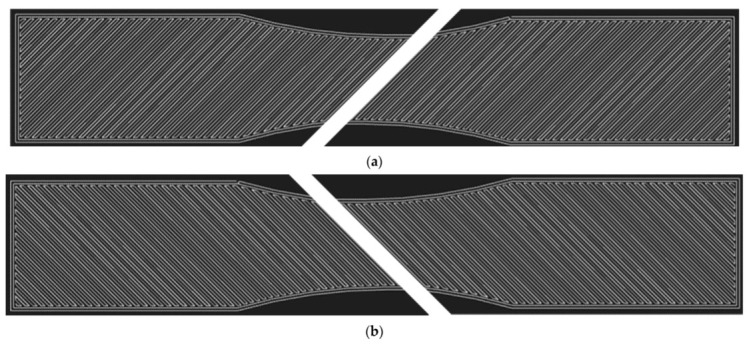
Infill orientation to loading axis: (**a**) 45° angle; (**b**) −45° angle. The layers are stacked alternately. The white lines represent the defined path of material deposition.

**Figure 7 materials-15-04224-f007:**
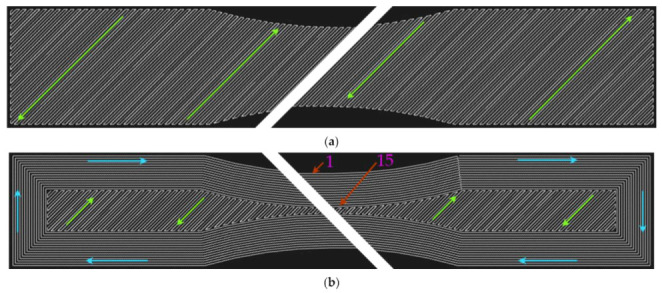
Infill pattern (green arrows) with an orientation of 45 degrees to the loading axis: (**a**) without walls; (**b**) with 15 walls (blue arrows) around the specimen circumference.

**Figure 8 materials-15-04224-f008:**
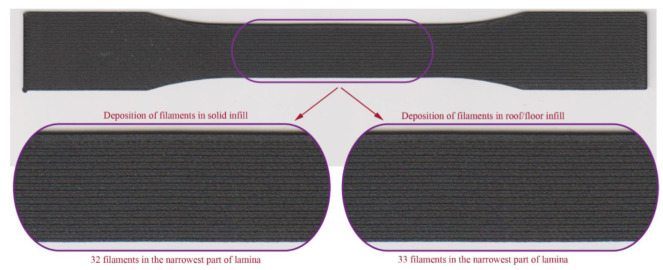
Deposition of filaments in the narrowest part of lamina.

**Figure 9 materials-15-04224-f009:**
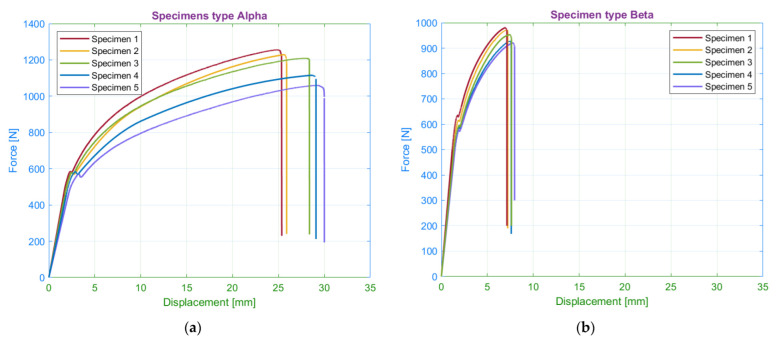
Tensile diagrams for specimens with infill orientation: (**a**) 45/−45; (**b**) 0/90; (**c**) 15/−75; (**d**) 30/−60.

**Figure 10 materials-15-04224-f010:**
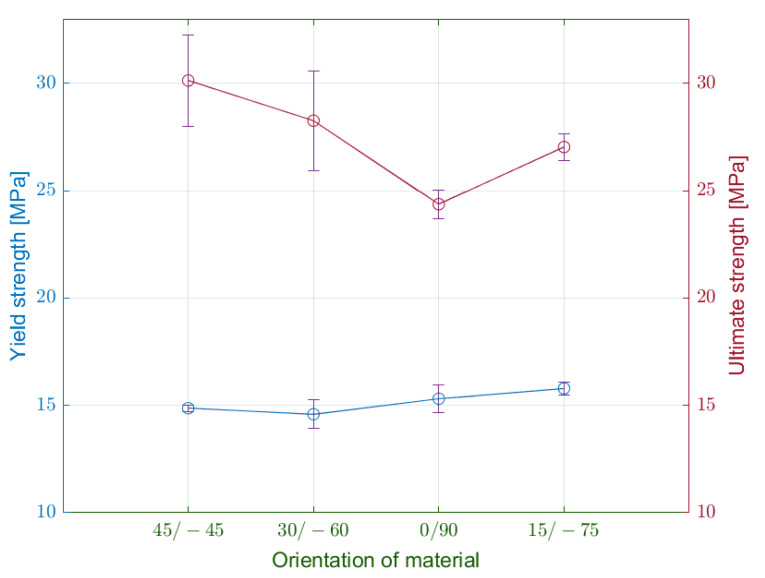
Comparison of measured average values of yield strength and ultimate strength for selected infill orientations.

**Figure 11 materials-15-04224-f011:**
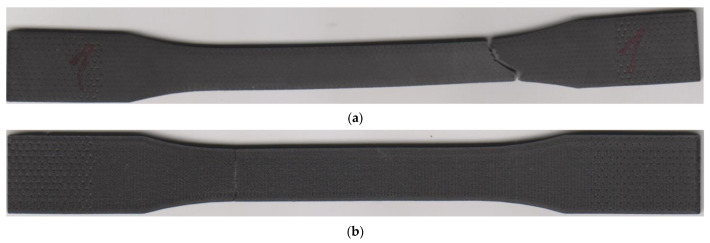
Fractured specimens: (**a**) infill orientation 45/−45; (**b**) infill orientation 0/90.

**Figure 12 materials-15-04224-f012:**
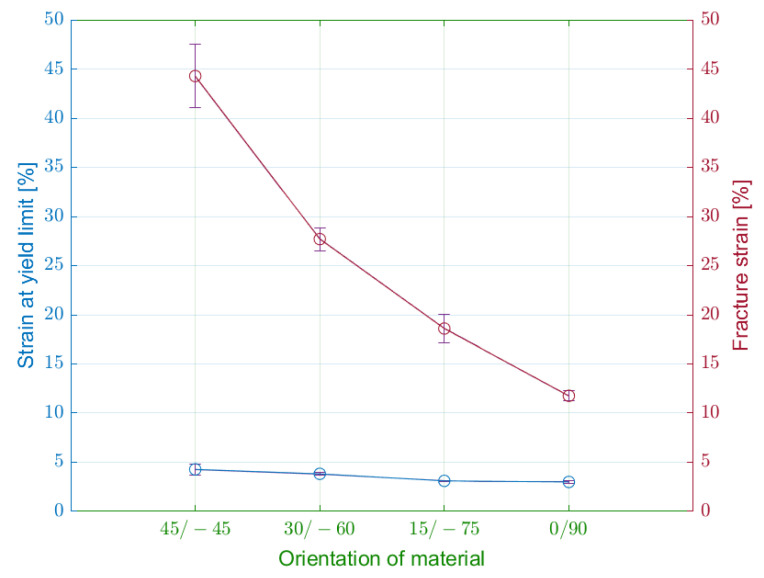
Influence of infill orientation on strain at the yield limit and fracture strain.

**Figure 13 materials-15-04224-f013:**
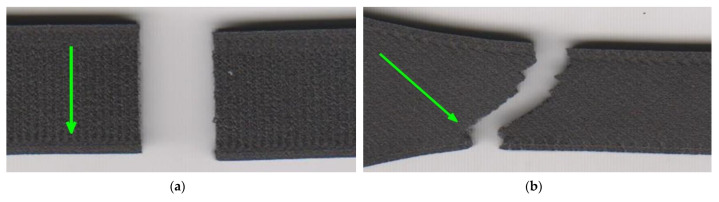
Detail of fracture: (**a**) 0/90 infill strategy; (**b**) 45/−45 infill strategy. Green arrows present the orientation of the solid infill in the top layer.

**Figure 14 materials-15-04224-f014:**
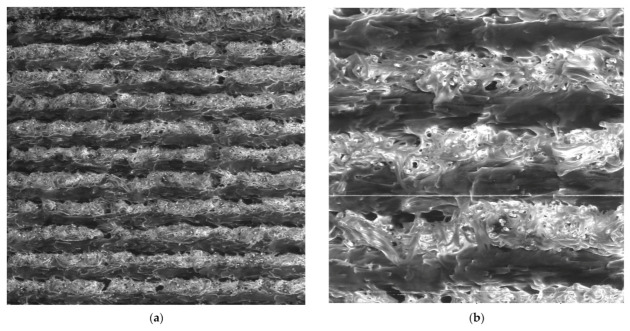
SEM images of fractured specimen alpha: (**a**) 100× magnification, detector LVSD; (**b**) 300× magnification, detector LVSD; (**c**) 300× magnification, detector BSE; (**d**) 500× magnification, detector BSE.

**Figure 15 materials-15-04224-f015:**
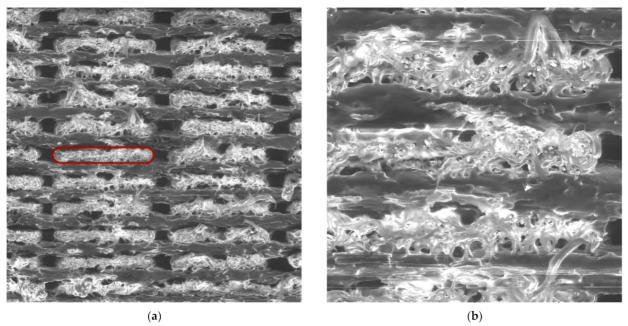
SEM images of fractured specimen beta: (**a**) 100× magnification, detector LVSD, red box presents one filament; (**b**) 300× magnification, detector LVSD; (**c**) 300× magnification, detector BSE; (**d**) 500× magnification, detector BSE.

**Figure 16 materials-15-04224-f016:**
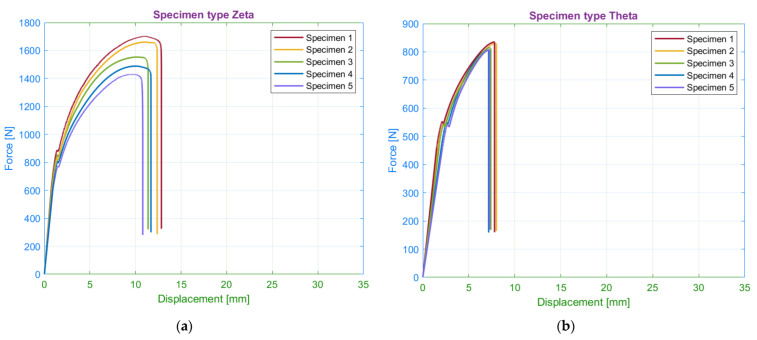
Tensile diagrams for specimens: (**a**) with 15 walls; (**b**) without walls.

**Figure 17 materials-15-04224-f017:**
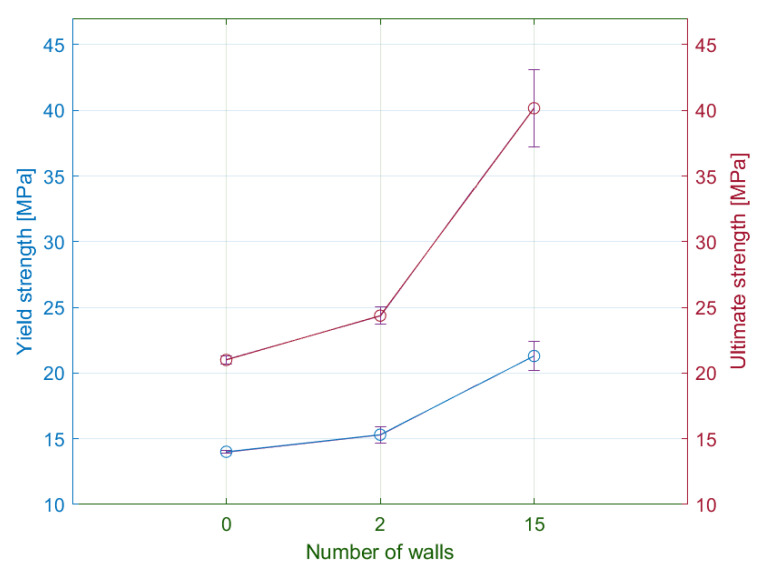
Comparison of measured average values of yield strength and ultimate strength for the selected wall number.

**Figure 18 materials-15-04224-f018:**
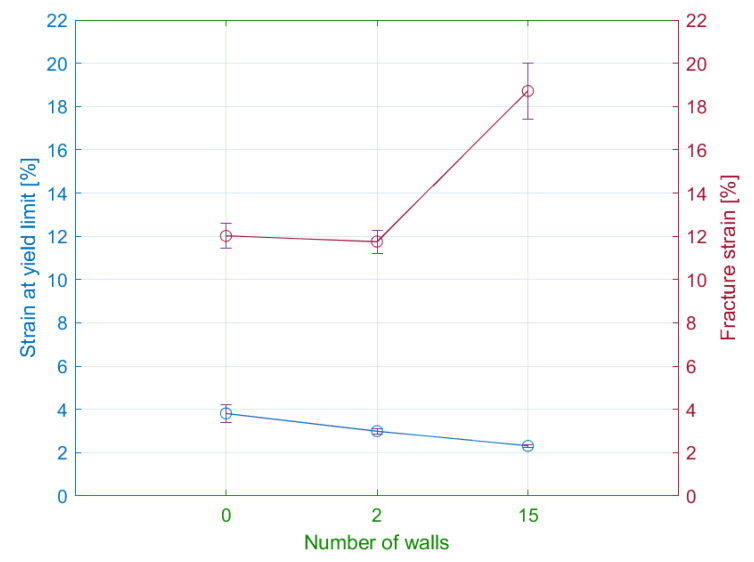
Influence of the number of walls on the strain at yield limit and fracture strain.

**Figure 19 materials-15-04224-f019:**
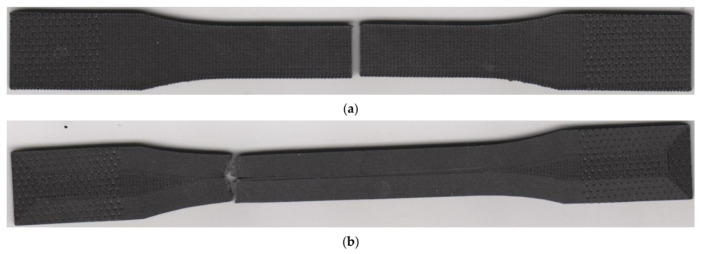
Fractured specimens: (**a**) without walls (theta series); (**b**) with 15 walls (zeta series).

**Figure 20 materials-15-04224-f020:**
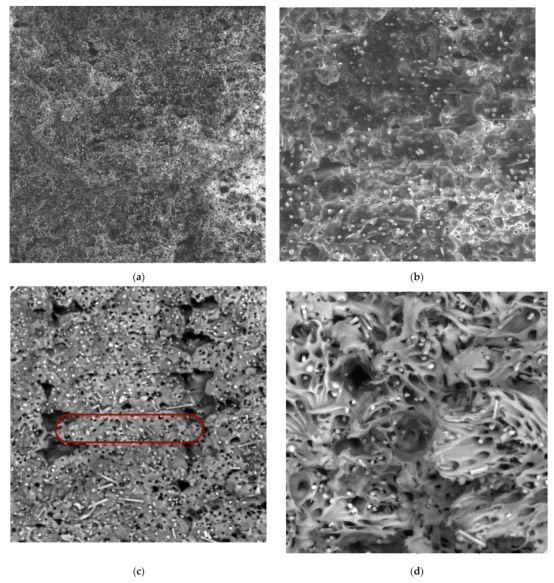
SEM images of fractured specimen zeta: (**a**) 100× magnification, detector LVSD; (**b**) 300× magnification, detector LVSD; (**c**) 300× magnification, detector BSE, the red box presents one filament; (**d**) 500× magnification, detector BSE.

**Figure 21 materials-15-04224-f021:**
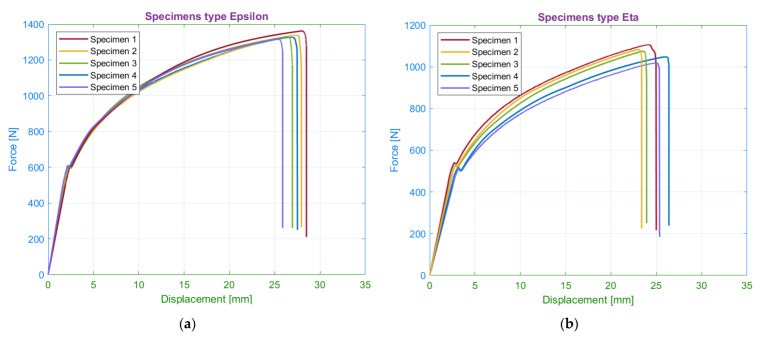
Tensile diagrams for specimens with (**a**) 10 roof/floor layers; (**b**) 0 roof/floor layers.

**Figure 22 materials-15-04224-f022:**
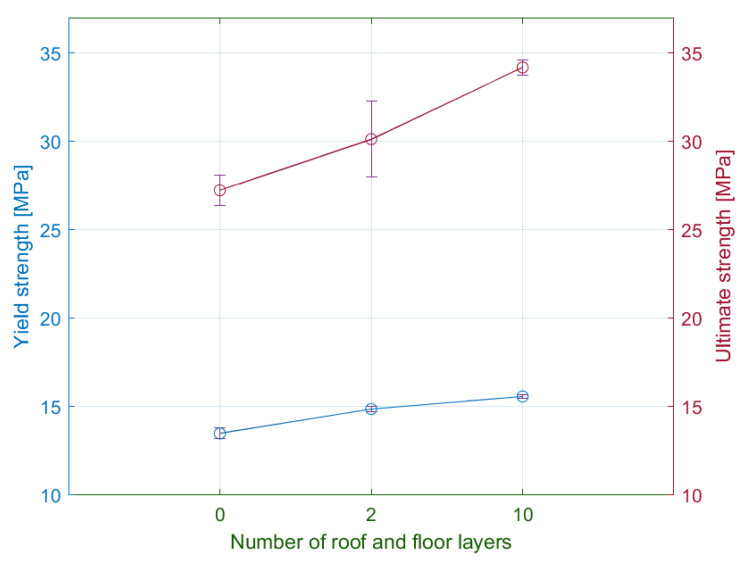
Comparison of measured average values of yield strength and ultimate strength for the selected number of floor/roof layers.

**Figure 23 materials-15-04224-f023:**
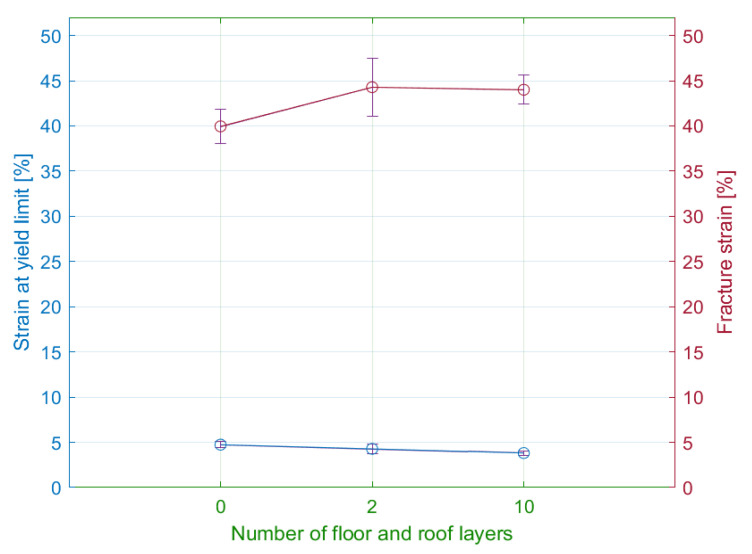
Influence of the number of walls on the strain for the yield limit and fracture strain.

**Figure 24 materials-15-04224-f024:**
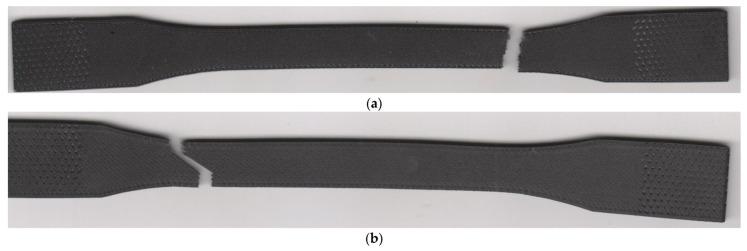
Fractured specimens: (**a**) 10 floor/roof laminas (epsilon series); (**b**) 0 floor/roof laminas (eta series).

**Figure 25 materials-15-04224-f025:**
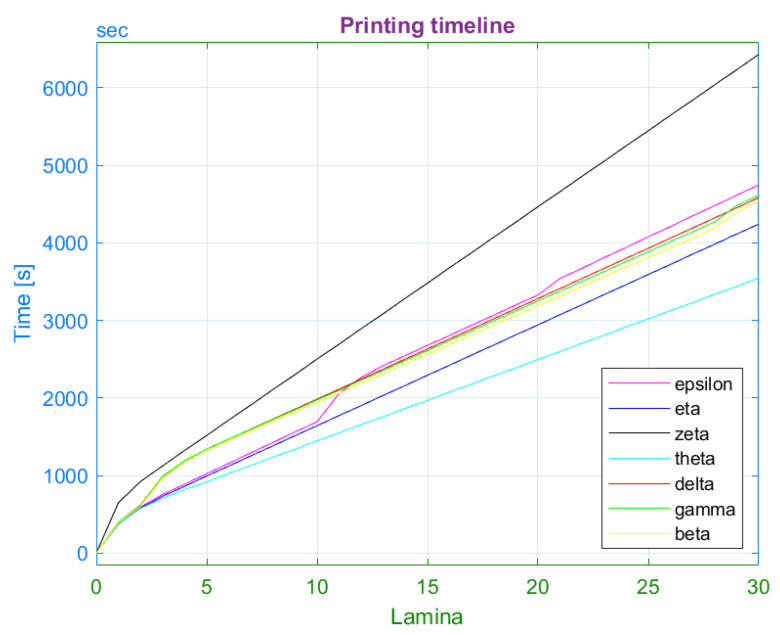
Printing timeline of each laminate series.

**Table 1 materials-15-04224-t001:** Mechanical properties of the material [29].

Parameter	Nylon Reinforced with Chopped Carbon Fibre (Onyx)
Tensile modulus of elasticity [GPa]	1.4
Ultimate strength [MPa]	30
Tensile strain at brake [%]	58
Printing method	FFF

**Table 2 materials-15-04224-t002:** Printing parameters.

Parameter	Value
Layer thickness [mm]	0.1
Base plane of specimen	XY
Matrix filament orientation in lamina [degrees]	0/90; 15/−75; 30/−60; 45/−45
Number of walls	0; 2; 15
Fill density [%]	100
Infill type	Solid fill
Number of floor and roof layers	0; 2; 10
Number of laminas	30

**Table 3 materials-15-04224-t003:** Results and specification of specimen series.

Series	Number of Walls	Infill Orientation	Number of Roof/Floor Layers	Yielding Force [N]	Maximum Force [N]	Printing Time [min]	Specimen Weight [g]
Alpha	2	45/−45	2	579.85	1175.203	83	8.48
Beta	2	0/90	2	597.28	950.62	81	8.3
Gamma	2	15/−75	2	615.91	1054.31	82	8.41
Delta	2	30/−60	2	569.25	1102.23	83	8.5
Epsilon	2	45/−45	10	607.82	1333.03	84	8.68
Zeta	15	0/90	2	830.78	1566.22	105	8.72
Eta	2	45/−45	0	526.05	1062.24	76	8.46
Theta	0	0/90	2	546.47	819.09	64	8.43

**Table 4 materials-15-04224-t004:** Elongation values for the selected infill orientations (mm).

Infill Orientation	Elongation at Yield Limit	Elongation at Fracture	Permanent Set
45/−45	5.4364	27.81	12.35
0/90	3.81	7.84	1.98
15/−75	3.95	11.68	3.46
30/−60	4.86	17.39	6.42

**Table 5 materials-15-04224-t005:** Elongation values for selected numbers of walls (mm).

Number of Walls	Elongation at Yield Limit	Elongation at Fracture	Permanent Set
15	2.94	11.75	4.44
0	4.85	7.55	2.47

**Table 6 materials-15-04224-t006:** Elongation values for the selected numbers of roof/floor layers (mm).

Number of Roof/Floor Layers	Elongation at Yield Limit	Elongation at Fracture	Permanent Set
0	6.025	20.09	10.87
10	4.85	27.63	11.86

**Table 7 materials-15-04224-t007:** Influence of infill orientation on specimen weight.

Infill Type	Reduction of Specimen Weight [%]	Reduction of Maximum Force [%]
45/−45	-	-
30/−60	0	−6.2
15/−75	−1	−10.3
0/90	−2.1	−19.1

**Table 8 materials-15-04224-t008:** Influence of selected printing parameters on the reduction of specimen weight and ultimate strength.

Number of Walls	Weight Reduction [%]	Reduction of Ultimate Strength [%]	Number of Floor/Roof Layers	Weight Reduction [%]	Reduction of Ultimate Strength [%]
15	-	-	10	-	-
0	−3.3	−47.7	0	−2.5	−20.3

**Table 9 materials-15-04224-t009:** Influence of the selected printing parameters on the reduction of printing time and ultimate strength.

Number of Walls	Printing Time Reduction [%]	Reduction of Ultimate Strength [%]	Number of Floor/Roof Layers	Printing Time Reduction [%]	Reduction of Ultimate Strength [%]
15	-	-	10	-	-
0	−2.9	−47.7	0	−2.1	−20.3

## Data Availability

Data is contained within the article.

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
