# Peer review of "Tensile Properties of Additively Manufactured Thermoplastic Composites Reinforced with Chopped Carbon Fibre"

_materials, 2022, doi:10.3390/ma15124224_

Round 1

Reviewer 1 Report

(1) This paper is lacking of novelty and  important coclusions;
(2) In all the figures, the standard should be added due to at least five specimens were measured;
(3) The manuscript looks like a testing report rather than a research paper;
(4) The motivation of this work should be clearly presented and attempt to solve some problems that remains to be unsolved in the previous literatures.
in surmmary, I would not like to recommond it for publication in this journal.

Author Response

   Dear Reviewer,

Thank You for the review of our submitted manuscript. Individual reviewers had various comments on some parts of the article (even contradictory). We have tried to make changes and improvements in the article based on the compromise and your comments.

Comments and Suggestions for Authors

Brief summary

Response to Broad comments

  1. This paper is lacking of novelty and  important coclusions;

Response:

The authors respect the reviewer's opinion. 3D printing is a relatively new technology whose technological principle of operation is based on the stacking of material layers on each other. In the case of the FFF method, the authors aimed to investigate the effect of filament orientation on tensile properties. The authors of the paper have no knowledge of any other papers dealing with a similar topic from a mechanics perspective. Similar articles published in peer-reviewed journals have focused almost exclusively on continuous reinforcing fibres, which report different behaviour in mechanical tests than chopped-fibre-reinforced nylon.

The conclusions from the measurements present new insights essential for the design of printed objects. In addition, they will be used in the FE calculation of a long-fibre reinforced composite (selection of a suitable material model and specification of selected characteristics), where the chopped-fibre reinforced nylon performs a matrix function.

  1. In all the figures, the standard should be added due to at least five specimens were measured;

Response:

The authors added standard deviations to the manuscript.

  1. The manuscript looks like a testing report rather than a research paper;

Response:

The aim of the article is to assess the influence of the orientation of nylon reinforced carbon fibre filaments on the tensile properties of the composite. The main part of the article is the evaluation of the tensile test, thanks to which the article can act as a testing report. The authors modified the article (mainly motivation) to emphasize the importance of assessing the tensile tests of composites.

  1. The motivation of this work should be clearly presented and attempt to solve some problems that remains to be unsolved in the previous literatures.

Response:

The authors have revised the introductory section describing the motivation of the work. The paper aims to assess the effect of material orientation on the tensile properties of the composite fabricated by the FFF method, which is composed of carbon fibre reinforced nylon. From a technological point of view, the FFF method works on the principle of controlled filaments deposition.

The motivation of the paper is to build on the findings realized on the same material with a solid infill material infill raster with varying material orientation in the structure for the following reasons:

  • Identify the effect of the orientation of the printed material with respect to the load axis on achieved tensile properties.
  • During the preparation process in slicing software, the user can change the printed object orientation on the printing desk (for example, the object does not fit on the printing desk due to its length). In the case of some printers, this modification means changing the orientation of the material arrangement in the structure. This act may inadvertently affect the resulting tensile properties of the printed composite by the user.
  • 3D printing is popular technology, which in practice means that printer users do not have to be specialists in fields like mechanics or materials but also members of the professional public who may not be aware of the impact of modifying the printing parameters on the tensile properties.
  • Throughout the lifetime of a printed object, the object may be loaded in various directions regarding the orientation of the material in the structure.
  • The material under investigation performs a matrix function for the CFRP composites, and knowledge of the properties of its constituents is essential for a proper understanding of its behaviour under various types of loadings.
  • In addition, modelling of the composites in finite element software requires the identification of the material parameters of the individual components and the selection of an appropriate material model.

Reviewer 2 Report

The authors are recommened to provide crossection images (SEM or optical/digital micrograph) of the broken tensile test specimens, which are important to evaluate the micro-structural change of the materials during tensile test.

Followings are some suggestion to improve the manuscript  

In the main text, definition of FFF (fused filament fabrication) method should be provided for the first time of its appearnce.

Page 2, line 66: "a long of short fibre", should be "long or shour fibres"

Page 3, line 115: "Article" should be "article"

The term "chapter" should be changed to "section", which is more appropriate for dividing parts of a research paper.

Page 7, line 197: "INSTRONT tensile machine", the series model of the machine should be provided.

Table 3, the name of specimen series should be the same with the ones mentioned in the text and figure. For example, series "A" should be series "alpha", series "B" should be series "beta"and so on.

Figure 13, the purpose of green arrows should be explained.

All graph figures: the title above each graphs can be removed, because it was mentioned in Figure's caption.

Conclusions: should be shortened (around 1/2 of the current length) and just focus on the main results/conclusion of the research. 

Author Response

   Dear Reviewer,

Thank You for the review of our submitted manuscript. Individual reviewers had various comments on some parts of the article (even contradictory). We have tried to make changes and improvements in the article based on the compromise and your comments.

Comments and Suggestions for Authors

Brief summary

Response to Broad comments

  1. The authors are recommened to provide crossection images (SEM or optical/digital micrograph) of the broken tensile test specimens, which are important to evaluate the micro-structural change of the materials during tensile test.

Response:

The comment was accepted. The authors have added to the text SEM crossection images of selected specimens.

  1. In the main text, definition of FFF (fused filament fabrication) method should be provided for the first time of its appearnce.

Response:

The authors have added an explanation of the abbreviation FFF at the point of first occurrence in the text.

  1. Page 2, line 66: "a long of short fibre", should be "long or shour fibres"

Response:

The authors have replaced "a long or short fibre" with the correct phrase "long or short fibres".

  1. Page 3, line 115: "Article" should be "article"

Response:

The error has been corrected by deleting the entire paragraph.

The comment was accepted in whole manuscript.

  1. The term "chapter" should be changed to "section", which is more appropriate for dividing parts of a research paper.

Response:

The comment has been accepted. The authors have replaced the term chapter with the term section.

  1. Page 7, line 197: "INSTRONT tensile machine", the series model of the machine should be provided.

Response:

The authors added the serial number of the tensile test device to the text -> Instron 5985.

  1. Table 3, the name of specimen series should be the same with the ones mentioned in the text and figure. For example, series "A" should be series "alpha", series "B" should be series "beta"and so on.

Response:

The difference occurred due to automatic correction of the text. The authors have unified the description of each series throughout the document.

  1. Figure 13, the purpose of green arrows should be explained.

Response:

The green arrows represent the direction of filament orientation. The authors accepted the comment and have added the purpose of the green arrows to the figure caption.

  1. All graph figures: the title above each graphs can be removed, because it was mentioned in Figure's caption.

Response:

The authors accepted the comment. The original graphs have been replaced by graphs without titles.

  1. Conclusions: should be shortened (around 1/2 of the current length) and just focus on the main results/conclusion of the research. 

Response:

The length of the conclusion was reduced approximately to 1/2 of its original length.

Reviewer 3 Report

Paper No.: Materials-1665369

Title:  

 Tensile properties of additively manufactured thermoplastic 2 composites reinforced with chopped carbon fibre

There are some issues that need to be treated.

Abstract

  • The introducing part in (line 9 to line 18) is too long. Please reduce it to 2 or 3 lines.
  • “The future modelling in FEA software requires additional experiments to verify the viscoelastic properties of the material”. It is not indicated in the manuscript how can these experiments serve the FEA?

Introduction

  • In line 35: “based on material subtraction ……”, please change “material subtraction” by the usual expression “material removal”.
  • In line 50: the abbreviation should be explained with the first mention “FFF method”.
  • Please move Figure 1 to the experimental part if it is your used method, else give a refence.
  • The authors have extensively presented certain process (with certain trade name). Please mention in summary the other methods.

Materials and Methods

  • The subsection (2.1) is a classic description of conducting the tensile test which is not directly serving the manuscript.
  • I suggest to describe material and the production method then evaluation techniques.
  • In line 149-150: Dimension tolerance of ±2mm is too much for a thickness of 3 mm. Please revise.
  • Is the number of walls the same as the number of laminas? It is better to present a schematic sketch of the sample and materials layer arrangement as that shown in the Figure 1.
  • Please use arrow and annotation to describe the samples structure in Figure 6 and other figures.
  • It is advised to gather the figures with similar content in one figure (such as Fig. 6 and Fig. 7).

Results

  • What is the reason behind appearing the yield point in such form (upper and lower yield points).
  • The authors have presented the effect of some production parameters (Orientation of infill pattern to the loading axis, Number of walls, Number of floor and roof layers) on the tensile stress-strain curves, the tensile yield, the ultimate tensile strength, the stains and the fracture behaviors. However, the same presentation way (curves and tables) has repeated for each production parameters: Orientation of infill pattern to the loading axis, Number of walls, Number of floor and roof layers.  

It would be more visible for the reader if the authors present the different effects on the same tensile property. For example make a subsection for presenting and discussing the effect of the different parameters on the yielding behavior, or on the fracture behavior of the material. In this way the reader can compare better.

Conclusions:

  • The conclusion is too long. Please concentrate on the new findings.

Author Response

   Dear Reviewer,

Thank You for the review of our submitted manuscript. Individual reviewers had various comments on some parts of the article (even contradictory). We have tried to make changes and improvements in the article based on the compromise and your comments.

Comments and Suggestions for Authors

Brief summary

There are some issues that need to be treated.

Response to Broad comments

  1. The introducing part in (line 9 to line 18) is too long. Please reduce it to 2 or 3 lines.

Response:

The comment was accepted. The authors have shortened the mentioned part of abstract to 3 lines.

  1. “The future modelling in FEA software requires additional experiments to verify the viscoelastic properties of the material”. It is not indicated in the manuscript how can these experiments serve the FEA?

Response:

Material properties are a necessary input for any modelling using finite-element software. The results published in this manuscript allow to perform finite element analysis of:

- Composites composed of nylon reinforced with chopped fibre,

- Composites reinforced with a long fibre, where the matrix function is performed by nylon reinforced with a chopped fibre.

The authors proposed a new modelling approach of CFRTP composites in FEA and these results of matrix allow us to choose the appropriate material model and to improve the accuracy of computations.

  1. In line 35: “based on material subtraction ……”, please change “material subtraction” by the usual expression “material removal”.

Response:

The authors accepted the comment. The term "material subtraction" was replaced by the term "material removal".

  1. In line 50: the abbreviation should be explained with the first mention “FFF method”.

Response:

The authors have added an explanation of the abbreviation FFF at the point of first occurrence in the text.

  1. Please move Figure 1 to the experimental part if it is your used method, else give a refence.

Response:

Figure 1 has been moved to the subchapter focused on the Manufacturing process. In addition, this new subchapter comprises information about the FFF printing method, which was initially part of the article introduction.

The purpose of Figure 1 is to introduce the fundamental parts of an FFF-based printer and to show the variability of material deposition opportunities that conventional manufacturing methods do not allow.

  1. The authors have extensively presented certain process (with certain trade name). Please mention in summary the other methods.

Response:

The authors modified the part of the introduction. A paragraph describing the essential 3D printing methods - extrusion, polymerization, lamination and sintering - has been added.

  1. The subsection (2.1) is a classic description of conducting the tensile test which is not directly serving the manuscript.

Response:

The subchapter containing description of the tensile test was removed from the manuscript.

  1. I suggest to describe material and the production method then evaluation techniques..

Response:

The authors accepted the comment. Firstly, the manuscript describes the production method (Fused Filament Fabrication) and the printed material, then the evaluation techniques of the results.

  1. In line 149-150: Dimension tolerance of ±2mm is too much for a thickness of 3 mm. Please revise.

Response:

The authors apologize for the mistake in writing.

Dimension tolerance is ±0.2 mm.

  1. Is the number of walls the same as the number of laminas? It is better to present a schematic sketch of the sample and materials layer arrangement as that shown in the Figure 1. 

Response:

No, the number of walls is not the same as the number of laminas. The authors accepted the comment and added a schematic sketch of the specimens to the text.

  1. Please use arrow and annotation to describe the samples structure in Figure 6 and other figures.

Response:

Authors accepted comment and have improved figures readability.

  1. It is advised to gather the figures with similar content in one figure (such as Fig. 6 and Fig. 7).

Response:

The comment was accepted. The figures were gathered in one figure.

  1. What is the reason behind appearing the yield point in such form (upper and lower yield points).

Response:

The mentioned composite consists of nylon and chopped carbon fibre. The tensile curve of the carbon fibre is linear. On the other hand, pure nylon shows similar tensile curves with upper and lower yield strengths. Thus, the reason for the occurrence of yield point may be a characteristic property of the nylon. The above assumption is confirmed by the results performed on composites reinforced with continuous fibre. At a lower fibre fraction (approximately 6%), it is possible to observe both an upper and a lower yield point. On the other hand, in the case of composites with a higher fibre volume fracture in the structure, the upper and lower yield strengths are not observable.

The authors took a deeper look at the measured values for the upper and lower yield points.

Series

Force [N]

Displacement [mm]

Upper yield

Bottom yield

Difference

Upper yield

Bottom yield

Difference

Alpha

579.91

567.27

12.64

5.43

5.74

0.31

Beta

602.99

595.64

7.35

3.83

4.01

0.18

Gamma

615.89

606.78

9.11

3.96

4.15

0.19

Delta

569.32

557.68

11.64

2.41

2.54

0.13

Zeta

830.79

821.23

9.56

1.45

1.52

0.07

The results show that the differences in measured forces between the individual series are not significant and may represent a measurement error. However, the difference in the measured displacements of the individual series is more noticeable - up to 5 times, which means that the orientation of the filaments influences the creep time.

  1. The authors have presented the effect of some production parameters (Orientation of infill pattern to the loading axis, Number of walls, Number of floor and roof layers) on the tensile stress-strain curves, the tensile yield, the ultimate tensile strength, the stains and the fracture behaviors. However, the same presentation way (curves and tables) has repeated for each production parameters: Orientation of infill pattern to the loading axis, Number of walls, Number of floor and roof layers.

It would be more visible for the reader if the authors present the different effects on the same tensile property. For example make a subsection for presenting and discussing the effect of the different parameters on the yielding behavior, or on the fracture behavior of the material. In this way the reader can compare better.

Response:

The authors have considered the reviewer's comment. The primary aim of the paper is to observe the effect of selected printing parameters on tensile properties. For this reason, during the manuscript preparation, the authors decided to subdivide the article according to the printing parameters, not mechanical properties.

  1. The conclusion is too long. Please concentrate on the new findings.

Response:

The length of the conclusion was reduced approximately to 1/2 of its original length.

Round 2

Reviewer 2 Report

Thank you for your response.

Reviewer 3 Report

The authors have treated the issues raised in the previous revision round.